# SOX2 and SOX21 in Lung Epithelial Differentiation and Repair

**DOI:** 10.3390/ijms232113064

**Published:** 2022-10-27

**Authors:** Evelien Eenjes, Dick Tibboel, Rene M. H. Wijnen, Johannes Marco Schnater, Robbert J. Rottier

**Affiliations:** 1Department of Pediatric Surgery, Erasmus MC-Sophia Children’s Hospital, P.O. Box 2040, 3000 CA Rotterdam, The Netherlands; 2Department of Cell Biology, Erasmus MC, P.O. Box 2040, 3000 CA Rotterdam, The Netherlands

**Keywords:** lung development, lung stem cells, regeneration, SOX proteins

## Abstract

The lung originates from the ventral foregut and develops into an intricate branched structure of airways, alveoli, vessels and support tissue. As the lung develops, cells become specified and differentiate into the various cell lineages. This process is controlled by specific transcription factors, such as the SRY-related HMG-box genes *SOX2* and *SOX21*, that are activated or repressed through intrinsic and extrinsic signals. Disturbances in any of these processes during the development of the lung may lead to various pediatric lung disorders, such as Congenital Diaphragmatic Hernia (CDH), Congenital Pulmonary Airway Malformation (CPAM) and Broncho-Pulmonary Dysplasia (BPD). Changes in the composition of the airways and the alveoli may result in reduced respiratory function and eventually lead to chronic lung disorders. In this concise review, we describe different intrinsic and extrinsic cellular processes required for proper differentiation of the epithelium during development and regeneration, and the influence of the microenvironment on this process with special focus on SOX2 and SOX21.

## 1. Introduction

The cells of the mammalian embryo become increasingly specialized and gradually lose their potency during development. The fertilized oocyte gives rise to all cells of the body as well as the extra-embryonic tissues, but during the next phases of development, the cells become lineage restricted and determined to differentiate into specialized cell types [1,2]. This pattern of development is also reflected during the formation of the lung, where multipotent epithelial progenitor cells eventually segregate into one of the lineages to become a mature epithelial cell [3]. Abnormalities in development or differentiation frequently leads to congenital and perinatal pulmonary anomalies, such as Congenital Diaphragmatic Hernia (CDH), Congenital Pulmonary Airway Malformation (CPAM) or Broncho-Pulmonary Dysplasia (BPD). This review is intended to provide an overview with focus on the roles of SOX2 and SOX21 in the epithelium and we refer to more elaborate reviews for more details [4,5,6,7,8,9].

## 2. SOX Proteins Mark Early Cell Fate-Determination

The earliest morphological sign of lung development is the appearance of a bud merging from the ventral side of the foregut. Until that moment, the expression of the SRY-related HMG-box 2 (*SOX2*) gene marks the foregut epithelium, whereas caudal-related homeobox 2 (*CDX2*) is expressed in the caudal region of the primitive gut. Immediate prior to the emergence of the lung bud, *SOX2* expression ceases and NK2 homeobox 1 (*NKX2-1*) becomes expressed in the primitive lung bud. After elongation of this primitive bud, the endodermal cells start to express *SOX2* again, but the distal part, which starts to bifurcate, starts expressing *SOX9*. Subsequently, the growing epithelial tubular structure starts to branch and the future bronchial tree expands through a process called branching morphogenesis. Thus, the growing tips are marked by the expression of SOX9, whereas the non-branching parts express SOX2. However, contrasting mouse lung development where SOX2 marks proximal epithelium and SOX9 distal epithelium, the SOX9+ tip cells in human lung consistently coexpress SOX2 at the pseudoglandular stage [10,11,12,13,14]. These tip cells loose SOX9 expression when they exit the branching tip of the epithelium.

The division between airway and alveolar fate is one of the earliest events in epithelial specification, and cells that become committed express either SOX2 or SOX9, respectively. Subsequently, intrinsic cellular changes occur as SOX2+ airway progenitor cells differentiate to airway specific cell types. Complete loss of SOX2 in airway epithelium does not result in branching defects, as shown previously [15,16], but it results in a loss of basal cells, decrease in cilia and secretory cells, during both development and regeneration [15]. On the other hand, ectopic expression of Sox2 in the developing lung epithelium resulted in enlarged airsacs lined with proximal epithelium, increased numbers of basal cells and pulmonary neuroendocrine cells (PNECs), resembling the CPAM phenotype [17]. In addition, the ectopic expression of Sox21 also led to enlarged airways, although smaller and less severe than the Sox2-induced airsacs [18]. Besides the importance of SOX2 in airway progenitor differentiation, SOX2 is critical for pluripotency in embryonic stem cells and development of many other organs, e.g., brain, esophagus and intestine [19,20,21]. Only in some progenitor cells, SOX2 is co-expressed with another SOX protein, SOX21, e.g., airway, inner ear, neuron, and embryonic stem cells [14,22,23,24,25]. Due to the wide spread role of SOX2, its function is subject of multiple studies, but the interaction of SOX2 with SOX21 on progenitor cell behavior has mostly been ignored. SOX2 and SOX21 have opposite effects on airway progenitor cell differentiation: SOX2 stimulates and SOX21 inhibits differentiation to fine-tune the balance in maintenance and turnover of airway progenitor cells [14]. Furthermore, deficiency in SOX2 or SOX21 showed to have opposite phenotypes in clustering PNECs during development [18]. Thus, SOX2 and SOX21 stimulate and repress sets of genes in an opposite manner to influence airway development.

## 3. The Level of Expression of SOX2 and SOX21 Determines Progenitor Cell Behavior

Common among stem-, progenitor-, and pluripotent- cells is a sensitivity to changes in SOX2 dosage [26,27]. A notable difference among these tissues is the effect of SOX2 expression levels on either proliferation or differentiation. Neural progenitor cells exit the cell cycle upon *SOX2* deletion [28], while trachea epithelium exhibit altered differentiation programs without changes in cell proliferation [14]. The regulatory function of SOX2 is highly dosage dependent, as was nicely illustrated by the regulation of cyclinD1 (CCND1) expression in neuronal progenitor cells. When expressed at low levels, SOX2 only binds to high-affinity sites to activate *Ccnd1*, whereas high levels allows SOX2 to also binds to low affinity sites to repress *Ccnd1* expression [28]. The levels of SOX2 and SOX21 increase during human and mouse adult basal cell differentiation in vitro, suggesting a dosage dependent regulatory function for both proteins [14]. Like SOX2, transcriptional targets of SOX21 might also be dosage dependent, and may interfere with SOX2-induced genes, due to its conserved DNA binding domain and potential binding to SOX2 [25,26].

The importance of SOX2 dosage is further shown by a defect in separation of trachea and esophagus early in development [21]. Upon separation, besides *SOX2*, *SOX21* was 7 fold increased in the esophagus compared to the trachea, suggesting that an early mechanism of SOX2 and SOX21 regulation of genes might be important in a correct separation [29]. However, *Sox21^−/−^* mice do not show a defect in separation of the esophagus and trachea. Furthermore, *Sox21^−/−^* mice show an increased differentiation of airway progenitor cells, but does not cause any respiratory distress. So, SOX21 may fine-tune SOX2 function, and is important in maintaining a progenitor state, but it probably does not regulate sets of genes that are essential for the development of the esophagus, trachea or airways. It would be valuable to identify the genomic targets of SOX2 and SOX21, and recently it was shown that these factors have unique and common genomic binding sites, but it may also be interesting to evaluate binding profiles of these factors at varying levels of expression [30].

## 4. Cofactors of SOX2 and SOX21 Determines Their Regulatory Function

SOX proteins alone do not elicit their action, but require binding to other (mostly) transcription factors that provide specificity [26,31,32]. A number of putative SOX2 binding partners were identified in E18.5 lung using a mouse model containing a biotinylated SOX2 [30,33]. The contradicting phenotypes of increased or decreased differentiation in lung progenitors upon loss of SOX2 or SOX21, respectively, could be in part explained by differences in binding partners [14,18]. SOX2 and SOX21 also interact with each other, and could as such differentially influence transcription [24,25]. Additionally, SOX2 and SOX21 have similar DNA binding motifs and bind similar co-factors, what would results in a competition in both binding to the DNA and/or to their co-factors.

Determining co-factors of SOX2 and SOX21, may be important to understand mechanisms underlying changes in progenitor function resulting in airway malformations, disturbed differentiation or aberrant repair. For example, patients with either AEG (Anaphtalmia-Esophageal-Genital)-syndrome or CHARGE (Coloboma, Heart malformations, Atresia of the choanae, Retarded growth, Genital anomalies and Ear anomalies)-syndrome are frequently misdiagnosed because of overlapping clinical conditions. However, AEG is associated with SOX2 and CHARGE is linked to the Chromodomain Helicase DNA Binding Protein 7 (CHD7) gene [34,35]. A potential molecular explanation of this phenotypic gradient displayed by AEG and CHARGE patients was provided by the discovery that SOX2 and CHD7 physically interact [31,36]. Moreover, several genes that are transactivated by this complex are associated with other, related syndromes, such as Jagged1 (JAG1; Alagille syndrome), MYC proto-oncogene N (MYCN; Feingold syndrome) and glioma-associated oncogene family zinc finger 3 (GLI3; Pallister-Hall syndrome). From the above example it can be hypothesized that changes in associating proteins may direct different responses guided by SOX proteins.

## 5. Micro-Environmental (Signaling) Interactions Guiding Differentiation and Regeneration

Cells undergo various differentiation, trans-differentiation and de-differentiation programs during development and regeneration which are characterized by changes in gene expression, as described for SOX2 and SOX21. These intrinsic changes are mostly influenced and triggered by the local tissue microenvironment. During development, different progenitors are present along the branching airways from proximal to distal and a subset of the adult progenitor cells remain in the epithelium at specific locations; submucosal glands, tracheal epithelium and neuroendocrine (NE) cells. Distinct niche components, including mesenchymal cells, neighboring epithelial cells, nerves, immune cells and extracellular matrix, can contribute to progenitor cell behavior, during development and from quiescence airway epithelium to active regeneration (Figure 1).

## 6. Mesenchymal-Epithelial Signaling Define the Extrapulmonary Airways

Distal mesenchymal Fibroblast Growth Factor 10 (FGF10) signaling inhibits differentiation of SOX9+ tip progenitor cells early in lung development, but once epithelial cells are committed to the SOX2+ airway epithelium, FGF10 can also induce basal cell differentiation [41]. SOX21 marks this proximal epithelial region where basal cell differentiation is induced and underlying FGF10 signaling is present [14]. In adult tracheal epithelium, basal cells maintain themselves and are recruited upon injury by mesenchymal expression of FGF10 (Figure 1A). The secretion of FGF10 is regulated by epithelial activation of the Hippo pathway effector, Yes1 Associated Transcriptional Regulator (YAP), leading to the expression and secretion of Wingless and Int1 7b (WNT7b) [42,43,44] (Figure 1A). YAP is essential for the maintenance of TRP63+ basal cells and loss of YAP results in increased differentiation to ciliated cells [43]. YAP regulates SOX2 expression in the transition from distal to proximal epithelium early in lung development [45]. Vice versa, SOX2 was shown to regulate transcription of YAP in cancer cells and osteo-adipo lineages [46,47]. Mutations in the *FGF10* gene are associated with an airway branching variation and with high risk of developing COPD [48].

## 7. NOTCH Signaling in Airway Epithelium Guides Differentiation

Notch signaling, is known for being expressed in a salt-and-pepper pattern, where receptor (NOTCH 1 to NOTCH4) and ligand (JAG1, JAG2, DLL1, DLL3 and DLL4) are expressed from neighboring cells and influence cell fate decisions during organ development [49]. Inhibition of the NOTCH signaling pathway, increases the differentiation to ciliated and NE cells at the expense of secretory cells (Figure 1B) [50]. The expression pattern of Jagged2 (JAG2) is initially in the most proximal airways around E12.5 and from E13.5 onwards it starts to overlap with the SOX2+ SOX21+ extrapulmonary airways, whereas JAG1 is only expressed in the SOX2+ intrapulmonary airways [14,51]. Within the inner ear and during neurogenesis, SOX2 and SOX21 is linked to NOTCH activity by regulating transcription of JAG1 and the NOTCH responsive gene hairy and enhancer of split-5 (HES5) [22,23,26,31]. SOX2 and SOX21 may regulate transcription of multiple NOTCH factors, such as JAG1 and JAG2, in extrapulmonary airways, as well as HES1 in the development of PNEC and non-PNEC cell fate. Furthermore, the increase in PNEC precursor cells upon the loss of SOX21 is similar to the phenotype observed by a loss of Distal-less (DLL) ligands [14,15,18,51]. This would suggest that both SOX2 and SOX21 do not only target a single NOTCH pathway, but may actually regulate multiple NOTCH and non- NOTCH signaling pathways dependent on airway specific cell type, which can be subject for future studies.

## 8. Neuronal Innervation Influences Progenitor Cell Differentiation

During the formation of the respiratory tract, an associated neural network is formed [52,53,54]. At the end of lung development, neurons innervate airway smooth muscle cells to control smooth muscle tone and trigger reflexes such as cough, but also innervates the SOX2+/SOX21+ airway epithelium, submucosal glands and PNECs (Figure 1C) [8,18,55,56,57]. Such a role between SOX2 expression and neuronal innervation has not yet been explored in the airway epithelium.

Defective innervation of respiratory epithelium has been associated with the development of Esophagus Atresia/Tracheo-Esophageal Fistula (EA/TEF) [58], as well as to branching defects in a CDH animal model [59]. Furthermore, disrupting neuronal innervation by laser ablation in explant cultures, resulted in a failure of local budding, suggesting involvement of neuronal innervation in the formation of the airway tree [60]. In the salivary gland, innervation has shown to be essential in maintaining the undifferentiated state of progenitor cells during development and regeneration [61,62]. A recent study showed that a subset of enteric glial cells regulated the regenerative capacity of resident stem cells in the intestinal crypt [63]. It would be interesting to evaluate the role of innervation on the regenerative capacity of the lung and the influences of SOX expression levels in innervated airway epithelium and PNECs. We showed that the first histological sign of innervation of NE cells occurs around E16.5 and corresponds with a dip in SOX2 and SOX21 protein levels [18,64]. Furthermore, we showed that reduced levels of Sox2 result in less PNECs/NEBs, whereas reduced levels of Sox21 showed the opposite effect. This indicates a delicate balance between these two transcription factors in generating NEBs. Moreover, this may also link to some pediatric lung diseases, such as CDH, which show increases in the number of PNECs [65].

## 9. Immune Cells Increase Complexity to Regulation of Airway Progenitor Behavior

In the lung, infiltration of immune cells often correlates with diseases. Macrophages are important in wound repair, and the lung contains at least three distinct populations of macrophages, each residing in specific niches: alveolar, interstitial and primitive macrophages [66]. The latter is the first to populate the fetal lung and may contribute to the development of alveoli and repair [66,67,68]. It is known that basal cells and PNECs, both innervated, can have a function within immune response and recruitment of immune cells [38,69,70]. PNECs secrete immune-modulatory peptides, and recently the spectrum of different combinations of neuropeptides and peptide hormones has been analyzed by scRNA seq of human and mouse PNECs [56]. We showed that the emergence of PNECs depends on the level of SOX2 and SOX21, which may also reflect on the species and number of secreted peptides [18]. In BPD, immune infiltration and NE cell hyperplasia is observed, but an interaction between neuronal innervation, immune response and onset of disease have yet been unexplored.

The infiltration of immune cells is a hallmark of tissue under stress, but it has also suggested that these cells serve as components driving stem cell behavior within homeostatic conditions [71]. Macrophages have been shown to support stem cell survival in the intestine and regulatory T cells are found near the stem cell niche of hair follicles and promote regeneration [72,73]. Upon exposure of inflammatory cytokine, IL-13, basal cells differentiate to goblet cells at the cost of ciliated cells [74,75]. However, the interaction between immune cells and airway progenitor cell differentiation under homeostatic conditions have yet been unexplored.

The complexity of different cell types and their interaction between immune cells and other cells during lung development was recently explored by single cell sequencing of complete mouse lung tissues at various time points. Subsequently, the study examined ligand-receptor pairs between cell types in a spatial-temporal manner to map cellular interactions during development [76]. Most, interactions were found within the immune compartment and sporadic interactions were found between the immune and non-immune compartment, which might include key signaling pathways for tissue development and homeostasis [76].

## 10. The Role of the Extracellular Matrix in the Airway Progenitor Niche

Extracellular matrix (ECM) components can play a key part in determining cell fate during development and regeneration (Figure 1D). The lung ECM continuously remodels during development [77], but the response of progenitor cells during development to ECM has not been well studied in the developing lung. The difference in dorsal-to-ventral localization of SOX2+SOX21+ airway epithelium early in lung development, as recently described, could be due to difference in ECM composition, besides differences in the mesenchymal population [14]. Such a key role for cell-ECM interactions was demonstrated in the developing pancreas, where progenitors that encounter a laminin- or collagen- enriched environment results in endocrine differentiation, while progenitors exposed to a fibronectin- or vitronectin- enriched ECM maintain their progenitor state [78].

Several chronic respiratory diseases, including idiopathic pulmonary fibrosis, asthma, and COPD, are associated with aberrant or excessive composition of ECM proteins. Besides immune infiltration, BPD also shows decreased vascularization, simplification of the alveolar compartment and a discomposed ECM [77,79]. A key role was demonstrated for Transforming Growth Factor-β (TGF-β) in the remodeling of the ECM in a BPD animal model, and basal cells exposed to TGF-β can be a source of aberrant ECM production [80,81]. Furthermore, production of matrix metalloproteinases by basal cells for remodeling the ECM showed to be essential for complete epithelial regeneration in vivo and differentiation towards the ciliated cell fate [82]. Studying the interaction between ECM components and progenitor cell behavior can help define therapeutic targets, but can also improve in vitro airway models to guide differentiation or to exactly mimic the physiological conditions found in vivo.

In conclusion, the differentiation of lung cells is affected by extrinsic and intrinsic signals. Disturbance in either of these signals will lead to congenital or perinatal lung abnormalities.

## Figures and Tables

**Figure 1 ijms-23-13064-f001:**
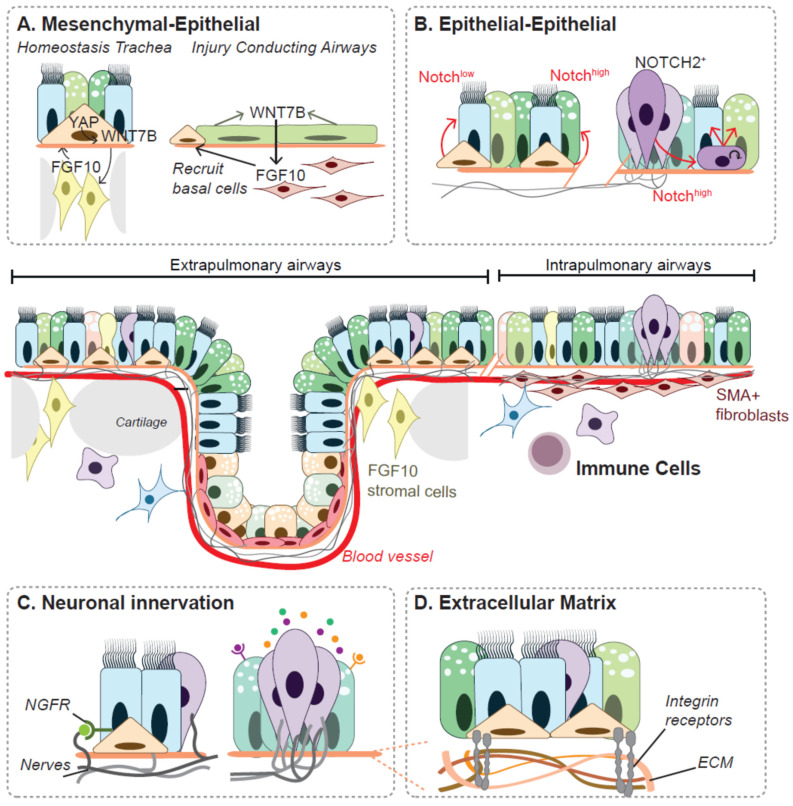
Extrinsic signaling cues coordinate lineage decisions of progenitor cells. Around SOX2+SOX21+ airway epithelium, trachea, submucosal glands and neuroendocrine cells, different cellular (mesenchymal-, epithelial-, neuronal- and immune- cells) and non-cellular components (extracellular matrix) contribute to the behavior of progenitor cells. (**A**) HIPPO-WNT-FGF reciprocal signaling between basal and underlying stromal cells, contribute to the maintenance of basal cells in the tracheal epithelium. Upon injury, damaged epithelium secrete WNT7B and FGF10 is secreted from Smooth muscle actin positive (SMA+) fibroblasts, which recruit basal cells to regenerate the epithelium. (**B**) Notch signaling guides differentiation of basal cells to secretory (Notch^high^) or ciliated (Notch^low^) cells. A subset of neuroendocrine (NE) cells in a NE cluster in Notch2^+^, and is suggested to function as stem cell. Upon injury, the NE dedifferentiate to a transit amplifying cells under control of high Notch signaling. This transit amplifying cell subsequently differentiates to ciliated or secretory cells, probably also through receiving high or low notch signal [37]. (**C**) The tracheal epithelium and neuroendocrine cells are innervated by neurons. Stimulation of PNECs can lead to the secretion of factors that might act on corresponding receptors present on the airway epithelium or can increase immune infiltrates [38,39,40]. Basal cells contain the P75-neuronal growth factor receptor (NGFR), which, potentially is stimulated by secretion of NGF by neuronal cells. However, this pathways has not yet been studied in the airway epithelium (**D**) Extracellular matrix (ECM) proteins are underlying the airway epithelial cells. Airway epithelial cells interact with ECM protein through integrin receptors, upon damage these interactions might change which can influence progenitor cell behavior.

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
