# Peer review of "SOX2 and SOX21 in Lung Epithelial Differentiation and Repair"

_ijms, 2022, doi:10.3390/ijms232113064_

Round 1

Reviewer 1 Report

This is an excellent  short review focused on the role of SOX2 and SOX21 in lung development, injury-repair and in pulmonary diseases. The authors provide a broad but concise overview of how these transcription factors interact with other pathways in various contexts. 

A few suggestions for improvement are:

 A sentence in the Introduction is missing to clearly state what will be focus of the review. The  current last sentence is somewhat vague, particularly when it refers to other” more elaborate reviews elsewhere”.

The authors refer to intrinsic and extrinsic cellular cues required for  cell fate decisions and differentiation. While extrinsic signaling cues are  well  discussed and illustrated in Fig. 1, it is less  clear what the intrinsic are.  Please clarify.  

 Please incorporate and discuss a recent report showing that  targeted disruption of Sox2 in the early lung endoderm does not influence  lung epithelial morphogenesis (Xie et al. Dev Cell 2021).

 Some of the sections are only peripherally related to Sox2-Sox21 (such as that about the role of ECM). Another section, describing the regulation of airway progenitors by immune cells, does not seem to have any clear connection with Sox2-Sox21. This can be further improved.  

Reviewer 2 Report

Authors reviewed the role of SOX2 and SOX21 in lung epithelial differentiation and repair by overviewing associated factors with progenitor cells. At each section, critical evidence was simply summarized, and the contribution of SOX2 and SOX21 to the development of lung can be understood from this review. However, minor issues should be addressed for further improvement of the manuscript as described in below.

1) At the sections regarding neuronal innervation and immune cells, there is no statement of SOX2 and SOX21. If no direct evidence to connect each section theme with SOX2/21, the contribution or the role of SOX2/21 in airway epithelium during each event should be stated or speculated.

2) It can be understood that SOX2 and SOX21 stimulate and repress sets of genes in an opposite manner to influence airway development from the section 2. “SOX proteins mark early cell fate-determination”. On the other hand, the relationship is not futured in following section, which would make us confused.

3) How about the transdifferentiation of type II alveolar epithelial cells into the type I. The involvement of SOX2/21 in the transdifferentiation is different from the case of early development of lung? The title would cover the alveolar epithelial cells.

4) Regarding CDH, CPAM, and BPD as described in Abstract, the involvement of SOX2/21 in these disorders should be deeply discussed. The significance of SOX2/21 cannot easily recognized without the evidence concerning the interaction of SOX2/21 with the issued disorders.

Round 2

Reviewer 2 Report

Author's comments and revision seems to be fine.